ⓐ | **Open Peer Review** | Clinical Microbiology | Research Article

# Application of ddPCR for diagnosis and treatment monitoring of visceral leishmaniasis patients: addition of an ultrasensitive tool to the diagnostic arsenal

Md. Abu Rahat,[1] Prakash Ghosh,[1,2,3] Rajashree Chowdhury,[1] Ashik Sharfaraz,[1] Fariha Tasneem Gawhar,[1] Md. Sakib Chowdhury,[1] Faria Hossain,[1] Nishad Tasnim Mithila,[1] Mostafa Kamal,[1] Md. Arko Ayon Chowdhury,[1] Priyanka Barua,[4] Rea Maja Kobialka,[2] Arianna Ceruti,[2] Ahmed Abd El Wahed,[2] Manfred Weidmann,[2] Malcolm Duthie,[5] Dinesh Mondal[1]

**ABSTRACT** Since the available tools in the arsenal toward accurate diagnosis and treatment monitoring of visceral leishmaniasis (VL) are limited. Therefore, efforts are still ongoing to develop an ultrasensitive molecular diagnostic tool applicable for endemic settings. In this evaluation study, we report on the efficiency of a digital droplet PCR (ddPCR) assay for diagnosis and treatment monitoring of VL. A total of 60 blood samples from VL patients were collected before the administration of antileishmanial drug. Upon treatment, blood samples were collected from each patient at 30 days and 180 days post-treatment. Blood samples from an equal number of healthy control subjects were collected from the same endemic region. Both qPCR and ddPCR were performed with the DNA extracted from blood samples to detect and quantify *Leishmania* DNA. Both of the molecular methods exhibited high sensitivity in diagnosing VL patients. Moreover, leishmania DNA was detected 180 days post-treatment in three VL cases that were diagnosed as VL relapse. The ddPCR showed an excellent analytical sensitivity in detecting as few as 0.1 *Leishmania donovani* parasite genome equivalents/reaction. With this promising quantitative ability, the ddPCR method showed a high positive correlation ($r = 0.8$) with the qPCR assay in determining the parasite loads in the clinical samples. Along with the commendable sensitivity, ddPCR showed absolute specificity as well. The findings of the study substantiate the excellent performance of ddPCR for diagnosing and treatment monitoring of VL patients. However, further multicenter studies are a prerequisite to evaluate the applicability of this tool in the endemic settings.

**IMPORTANCE** This is the very first study to report the use of digital droplet PCR (ddPCR) for diagnosis and treatment monitoring of visceral leishmaniasis patients. By leveraging this ultrasensitive tool, absolute quantification of the parasite in the clinical samples is possible without requiring standard DNA materials. Eventually, this method will be useful for monitoring the relapse of visceral leishmaniasis (VL) patients and the post-kala-azar dermal leishmaniasis patients as well, who have been playing a critical role in transmitting the disease at the post-elimination period of VL. Moreover, the successful establishment of this method will pave the way for the detection and quantification of the parasite in asymptomatic carriers and sand fly vectors, which will strengthen the post-elimination surveillance of visceral leishmaniasis in the endemic settings.

**KEYWORDS** visceral leishmaniasis, diagnosis, ddPCR, treatment monitoring

Leishmaniasis is an anthroponotic vector-borne disease caused by over 20 *Leishmania* parasite species, with the parasite transmission occurring via bites of infected female sandflies into human skin (1). The disease manifests in three main forms, and the most

**Peer Reviewers** Ariful Basher, Mymensingh Medical College and Hospital, Mymensingh, Bangladesh; Alejandro Schijman, Instituto de Ingeniería Genética y Biología Molecular, Buenos Aires, Argentina

Address correspondence to Prakash Ghosh, prakash.ghosh@uni-leipzig.de.

Md. Abu Rahat, Prakash Ghosh, and Rajashree Chowdhury contributed equally to this article. Author order was determined by drawing straws.

The authors declare no conflict of interest.

severe form, visceral leishmaniasis (VL, also known as kala-azar), affects the internal organs, particularly the spleen, liver, and bone marrow. Without proper treatment, VL can be fatal. Over 600 million people are at risk of VL worldwide, where Bangladesh, India, and Nepal together account for approximately 30% of the global burden of visceral leishmaniasis (2, 3). Considering the crucial need for targeted interventions and coordinated efforts to combat the spread of this endemic disease on the Indian subcontinent, a Kala-Azar Elimination Program (KAEP) was incepted in 2005 to eliminate visceral leishmaniasis. In line with the vision of KAEP, the Bangladesh government started the National Kala Azar Elimination Program in 2008 under the supervision of the Ministry of Health and Family Welfare (4). With the consistent and coordinated efforts, Bangladesh achieved the official validation from WHO in 2023 as the very first country to eliminate VL as a public health problem (5). Despite elimination, as for any vector-borne disease, if the reservoirs of infection remain, recrudescence of the disease is very likely.

On the Indian subcontinent, VL incidence follows a cyclic pattern with 5–10 year periods of high incidence, followed by 10–20 years of low incidence (6). Preventing the reemergence of the disease is a challenge, especially due to asymptomatic individuals who carry residual infection and eventually transmit the parasite in the community. It is observed that around an index case, while several individuals were infected but remained asymptomatic, a significant proportion of these individuals developed kala-azar (7). Likewise, VL relapse cases can contribute to parasite transmission (8). Furthermore, a significant proportion of the treated VL cases that develop Post-Kala-azar Dermal Leishmaniasis (PKDL) play a critical role in transmitting the disease during inter-epidemic periods. In the post-elimination era, it is expected that relapse and sporadic new cases of VL will still occur in endemic regions, albeit at a lower frequency. Therefore, because there is a high chance of resurgence of disease in previously endemic and low-endemic areas, a robust surveillance system is mandatory to maintain control.

Because of the reemergence of the disease from residual infections, Nepal, for example, has not yet received validation of VL elimination as a public health problem (9). A study demonstrated that 573 villages of Bihar, India, reported VL cases in 2018 despite not having reported kala-azar in 2013–2017, and these cases contributed to 20% of the total burden of kala-azar in the state (10). Additionally, an outbreak was spotted in 2018 in Kosra village, Bihar, India, which was once considered a low-endemic area for kala-azar (11). These examples indicate the importance of preventing the potential transmission and resurgence of the disease in maintaining the eliminated state in Bangladesh.

To track down the transmission and residual infection among the community, periodic surveillance is imperative during the post-elimination period to achieve the zero transmission goal. Toward achieving this ambition and accelerating surveillance activities, an ultrasensitive diagnostic method gauging the residual infection is indispensable.

The rk39 rapid detection test (RDT) for the detection of VL cases, widely used at present, is likely to lose its positive predictive value because of the low number of VL cases in the post-elimination phase, posing a challenge for accurate diagnosis of the disease (12). Furthermore, the rK39 RDT also exhibits low sensitivity in cases of co-infection, particularly in individuals co-infected with HIV (13). The advancement of molecular methods such as quantitative PCR (qPCR) overcomes the limitations of serology-based methods. However, the need for costly reference materials, variation in results across different laboratories, assays, and operators due to relative quantification methodology, and false negative results due to the effects of inhibitors might limit the use of qPCR in the post-elimination era.

With a view to overcoming the limitations of qPCR, digital PCR (dPCR) has been developed as an ultrasensitive detection tool. With this advanced technique, absolute quantification of the DNA amounts in a sample is possible while dependence on reference standards is not required. Moreover, this ultrasensitive technique can detect even at the single-molecule level (14). Considering these multifarious advantages, over the past years, digital droplet PCR (ddPCR) on the QX200 Droplet Digital PCR system

(Bio-Rad) has steadily become the most widely used ddPCR platform across various fields (15).

The ddPCR technology is rapidly replacing real-time qPCR as an efficient method for DNA quantification without the reliance on standards or external calibration and has been shown to be slightly more robust, especially for low target concentrations (16). The method is more resistant to PCR inhibitors, expunging the need for using technical replicates, which further increases the efficacy of ddPCR over qPCR (17, 18). Numerous studies have demonstrated a higher analytical/diagnostic sensitivity of ddPCR than qPCR in diagnosing disease, monitoring minimal residual infection, and disease progression (19–21).

Considering the superior quantitative ability of ddPCR, it can theoretically be exploited and implemented as part of the surveillance strategy of residual infection in the post-elimination era. However, before implementing such a tool, it is imperative to know the efficiency of the assay in detecting clinical cases. Therefore, in this study, we investigated the performance of a ddPCR assay for the diagnosis of VL patients and monitoring of their treatment.

## RESULTS

### Study participants

The demographic and clinical details of the participants from whom the samples were collected are listed in Table 1. Among 60 clinically confirmed VL patients, 11 (18.33%) patients had a previous history of VL, and none of the patients had any PKDL history in their lifespan. Thirty-six (60%) of the study-enrolled cases were male. The mean age of VL and endemic healthy controls was 28.7 (SD: 16.05) and 30.2 (SD: 13.63), respectively. Among the VL patients, 59 cases exhibited splenomegaly for less than 2 weeks, while six of the cases had hepatomegaly. All of the VL patients enrolled in the study were responsive to the treatment and declared clinically cured after 6 months of treatment. However, three of them reported a relapse within 1 year of completing treatment.

### Performance of ddPCR assay

#### *Linearity and limit of detection*

The linearity of the ddPCR assay was determined by quantifying 10-fold serial dilutions (with seven individual points) of a known parasite DNA corresponding to 10,000 to 0.01 parasite genome equivalents/reaction (22). More than 99.9% of droplets were saturated at a dilution of $10^{-4}$ (Fig. 1A and B). The standard curve generated was linear along the six-dilution step range with a correlation coefficient ($R^2$) of 0.99 ($P < 0.0005$) (Fig. 2). Signal was not detected in any of the negative controls. The assay detected as low as 0.1 *Leishmania donovani* parasite genome equivalents/reaction, equivalent to 10 fg of *L. donovani* genomic DNA.

### Reproducibility

Coefficient of variation (CV) was used to assess assay reproducibility. To calculate intra- and inter-assay CV, the copies per reaction quantified by the assay in the six 10-fold dilution series ($10^4$–$10^{-1}$) and clinical samples were converted to logarithmic values using a $\log_{10}(x+1)$ conversion. The intra-assay CV of the copy number between replicates was 1.15% for $10^4$, 0.47% for $10^3$, 0.31% for $10^2$, 1.72% for $10^1$, 1.53% for $10^0$, and 87.5% for $10^{-1}$ (Table 2). The inter-assay CV of the copy number between the three independent assays was 0.76% for $10^4$, 0.47% for $10^3$, 0.31% for $10^2$, 0.92% for $10^1$, 7.41% for $10^0$, and 100% for $10^{-1}$ and 3.08%, 1.02%, 7.41%, 11%, 18.56%, and 0.68% for the six clinical samples, respectively (Table 2). The CV values demonstrate a high reproducibility in all but except samples those with very low DNA copies.

**TABLE 1** The demographic and clinical indices of the participants

| Variables | VL cases (*N* = 60) | Endemic controls (*N* = 60) |
|---|---|---|
| Sex | | |
| Male, n (%) | 36 (60%) | 31 (51.67%) |
| Female, n (%) | 24 (40%) | 29 (48.33%) |
| Age (years) | | |
| Mean (SD) | 28.7 (16.05) | 30.2 (13.63) |
| History of VL, n (%) | 11 (18.33%) | 0 (0%) |
| History of PKDL | 0 (0%) | 0 (0%) |
| Fever ≥14 days, n (%) | 59 (98.33%) | 0 (0%) |
| Splenomegaly, n (%) | 59 (98.33%) | 0 (0%) |
| Hepatomegaly, n (%) | 6 (10%) | 0 (0%) |
| Pancytopenia, n (%) | 46 (76.67%) | 0 (0%) |
| Anemia | 59 (98.33%) | 0 (0%) |
| Darkening of skin | 1 (1.67%) | 0 (0%) |
| Positive in Rk39 RDT, n (%) | 60 (100%) | 0 (0%) |

## Sensitivity and specificity of ddPCR and qPCR assays

The sensitivity of the ddPCR assay was found to be 100% (CI: 94.04–100.00), and no false positive results were observed among the 60 negative controls (endemic healthy controls [EHC]). The ddPCR assay quantified DNA copies with a mean value ($\log_{10}$) of 1.53 (SD: 0.74) copies/reaction. The sensitivity and specificity of qPCR were both found to be 100%. The mean($\log_{10}$) parasite load quantified by the qPCR assay was 0.8 (SD: 0.86) parasite genome equivalents/reaction.

## Comparison between ddPCR and qPCR assay

The ddPCR showed absolute concordance with the qPCR assay (kappa: 1, McNemar *P* = 1.00) in detecting *L. donovani*. To compare the parasite burden between ddPCR and qPCR, we converted the values obtained from the qPCR assay to absolute copy number by multiplying the parasite load by 10, 50, and 100, and converted them to logarithmic values using $\log_{10}$ conversion. In the case of the first conversion (multiplying by 10), the Bland-Altman analysis demonstrates an excellent agreement (ICC = 0.8) between ddPCR and qPCR assay (Fig. 3A). Almost all the observations lie within the 95% limit of agreement with a mean difference of 0.06. The other two conversions (50 and 100) yielded a relatively lower agreement between the two assays with ICC values of 0.5 and 0.4 and a higher mean difference of 0.76 and 1.06, respectively (Fig. 3B and C). Moreover,

**TABLE 2** Repeatability and reproducibility of the ddPCR assay

| Parasite load | Inter-assay variation of copies per reaction | | | | | | Intra-assay variation of copies per reaction | | | | | |
|---|---|---|---|---|---|---|---|---|---|---|---|---|
| | Assay 1 | Assay 2 | Assay 3 | Mean | SD | CV | Replicate 1 | Replicate 2 | Replicate 3 | Mean | SD | CV |
| $1 \times 10^4$ | 172,000 | 194,000 | 166,400 | 177,466.67 | 14,589.49 | 8.22 | 174,000 | 149,800 | 194,000 | 172,600 | 22,133.23 | 12.82 |
| $1 \times 10^3$ | 17,620 | 16,280 | 17,880 | 17,260 | 858.6 | 4.97 | 17,160 | 17,840 | 16,780 | 17,260 | 537.03 | 3.11 |
| $1 \times 10^2$ | 1,640 | 1,686 | 1,742 | 1,689.33 | 51.08 | 3.02 | 1,640 | 1,680 | 1,620 | 1,646.67 | 30.55 | 1.86 |
| $1 \times 10^1$ | 142 | 154 | 150 | 148.67 | 6.11 | 4.11 | 194 | 230 | 204 | 209.33 | 18.58 | 8.88 |
| $1 \times 10^0$ | 12 | 13.2 | 8.8 | 11.33 | 2.27 | 20.07 | 20 | 20 | 18 | 19.33 | 1.15 | 5.97 |
| $1 \times 10^{-1}$ | 0 | 3.6 | 1.2 | 1.6 | 1.83 | 114.56 | 0 | 2 | 2 | 1.33 | 1.15 | 86.6 |
| ID-01 | 76 | 100 | 90 | 88.67 | 12.06 | 13.6 | NA[a] | NA | NA | NA | NA | NA |
| ID-02 | 88 | 94 | 94 | 92 | 3.46 | 3.77 | NA | NA | NA | NA | NA | NA |
| ID-03 | 24.6 | 16.4 | 24 | 21.67 | 4.57 | 21.1 | NA | NA | NA | NA | NA | NA |
| ID-04 | 11.4 | 9.4 | 6.6 | 9.13 | 2.41 | 26.4 | NA | NA | NA | NA | NA | NA |
| ID-05 | 6.6 | 6.2 | 14 | 8.93 | 4.39 | 49.17 | NA | NA | NA | NA | NA | NA |
| ID-06 | 866 | 812 | 864 | 847.33 | 30.62 | 3.61 | NA | NA | NA | NA | NA | NA |

[a]NA, not applicable.

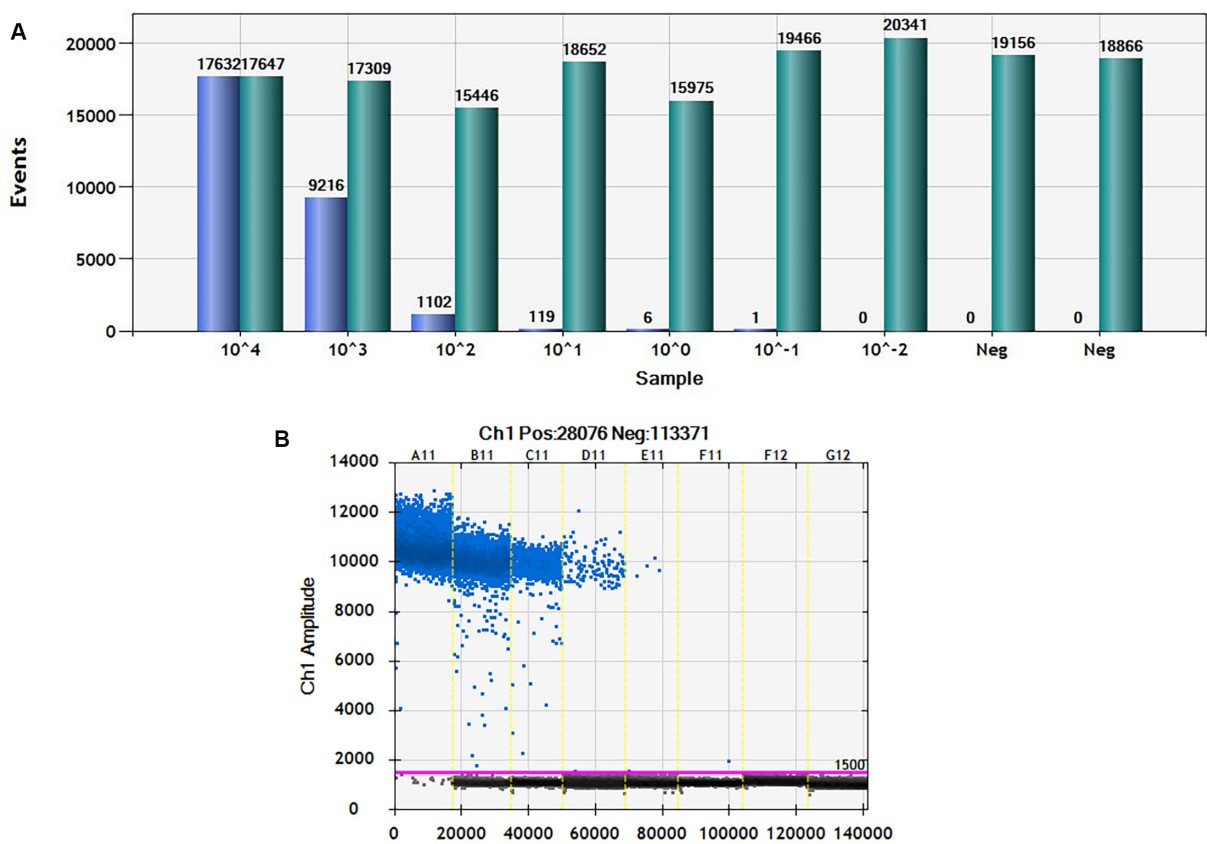

**FIG 1** Results from ddPCR experiments. (A) Representative bar graph of ddPCR analysis from blood DNA samples for identifying *Leishmania* DNA. Light green bars represent the total droplets number and blue bars represent the positive droplets. (B) One-dimensional scatter plots showing amplitude of fluorescence on the Y-axis and event number on the X-axis for the positive droplets.

a higher positive correlation ($R^2$ = 0.602) was observed between the two molecular assays in the quantification of the parasites (Fig. 4).

## The performance of molecular assays in disease monitoring

All 60 treated VL patients were monitored for 1 year after the completion of treatment and samples from 1-month time point (mtp) and 6-mtp were used to assess the prognostic value of ddPCR and qPCR assay (Fig. 5A and B). No DNA copies were detected by either of the assays in 1-mtp, where 5% (3/60) patients found to be positive by both the assays in 6-mtp with moderate parasite load had also developed symptoms. No *L. donovani* DNA copies were found in healthy controls with either of the assays at any time points.

## DISCUSSION

With the holistic efforts of NKEP, Bangladesh successfully maintained VL (Kala-azar) elimination status for three consecutive years during the validation phase and received the certification of elimination in 2023 (23). The current roadmap of the WHO in Bangladesh is to achieve zero transmission by 2025 and a Kala-azar-free status by 2030 (24). The current challenges in this post-elimination era include the limited tracking system for suspected KA cases from non-endemic regions and limited cross-border surveillance. As VL elimination has been achieved, efforts for active case detection of the NKEP are unlikely to be persistent due to the lack of resources. Besides, in the post-elimination period, the positive predictive value of rK39 RDT might decline due to the reduced prevalence of VL (12). To address these challenges, tailoring a disease

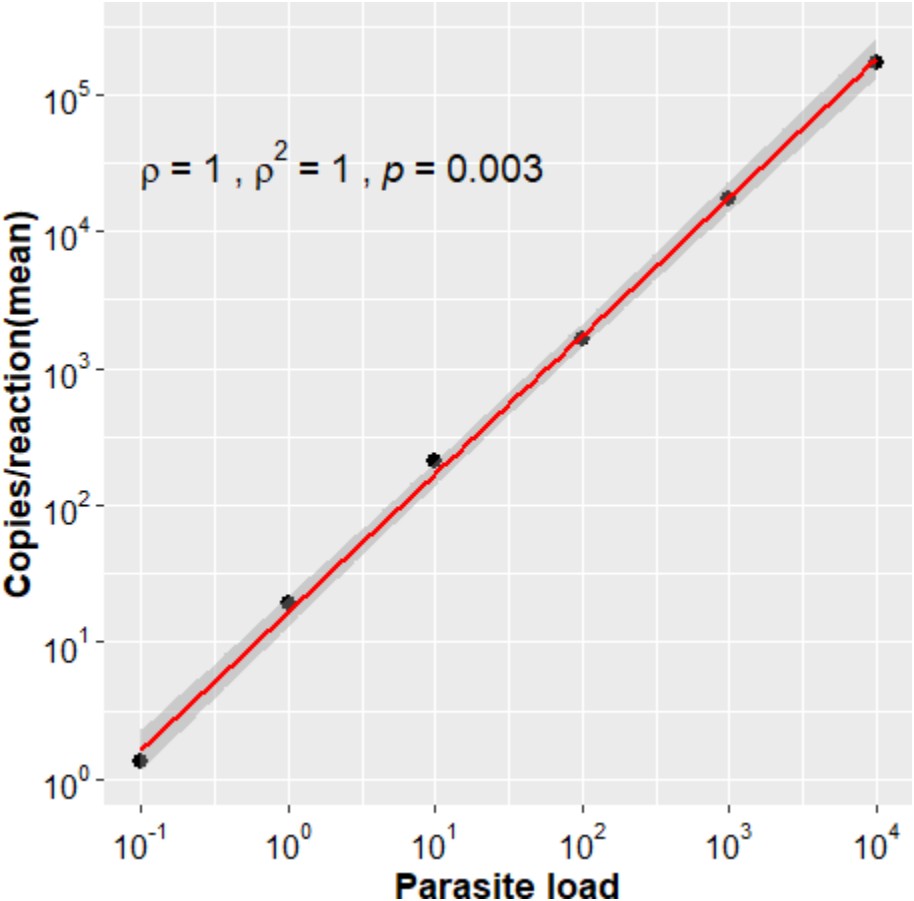

**FIG 2** The standard curve with a six-dilution range (10,000–0.1 parasite genome equivalents/reaction) exhibiting the limit of detection of ddPCR.

tracking system by incorporating an ultrasensitive surveillance tool has become the most important focus in the WHO agenda (25).

Owing to the limited funding for research on this neglected tropical disease, the existing arsenal to detect *Leishmania* infection accurately is limited. Considering the pitfalls of the serological methods, a handful of molecular tests have been developed to diagnose VL in endemic countries (26, 27). However, standardization or harmonization of the molecular methods is still required in different settings prior to broader application. To date, an RPA assay is the only test that has been widely evaluated on the Indian subcontinent for the detection of *Leishmania* DNA in clinical samples (28). Further studies are ongoing to exploit the quantitative ability of this point-of-need for treatment monitoring (personal communication). In the reference settings of the endemic countries, the PCR-based molecular methods have streamlined the diagnosis of VL, and thus, proper treatment of the patients has been ensured (29). Importantly, the diagnosis of relapse VL has become accurate, and the treatment monitoring has become possible. In addition to diagnosis, host surveillance has become cardinal in detecting the residual infection at peri- and post-elimination of VL (30). In this period, an ultra-sensitive tool is indispensable to monitor the ongoing transmission (30, 31). In this study, a third-generation molecular tool was evaluated for the diagnosis of VL.

The most promising findings of the study are the absolute sensitivity and specificity of ddPCR in detecting clinically confirmed VL cases. Congruent with our present findings, Ramírez et al. have reported 100% sensitivity and specificity of ddPCR assay in detecting *Trypanosoma cruzi* infections (32). Furthermore, Cheng et al. demonstrated 100% sensitivity and specificity of ddPCR assay for detecting Mycobacterium in

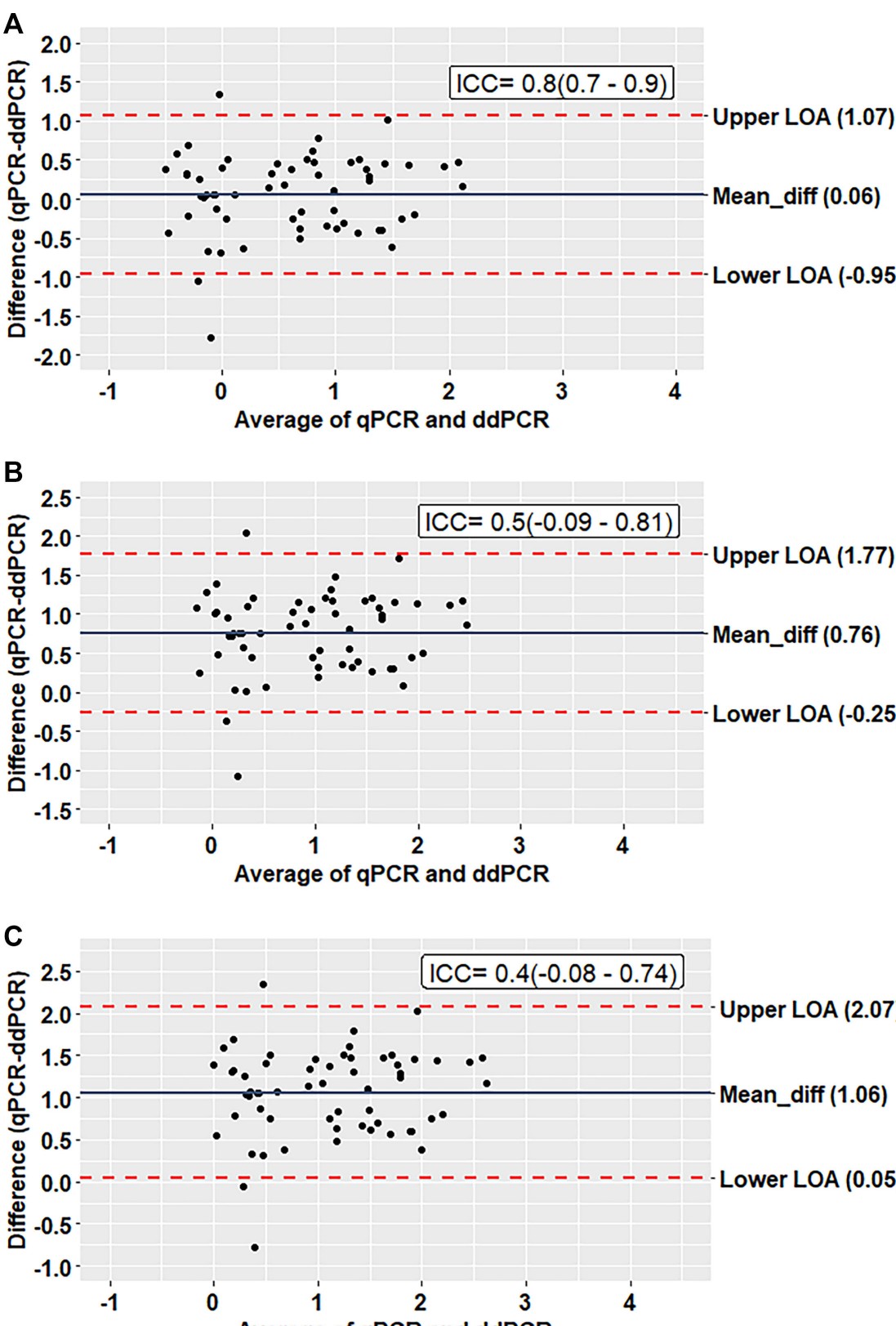

FIG 3   Bland-Altman analysis to determine the agreement between qPCR and ddPCR assay (unit = copies/uL). (A) Assuming 10 copies present in one parasite and converting the value to ln(copies/uL). (B) Assuming 50 copies present in one parasite and converting the value to ln(copies/uL). (C) Assuming 100 copies present in one parasite and converting the value to ln(copies/uL).

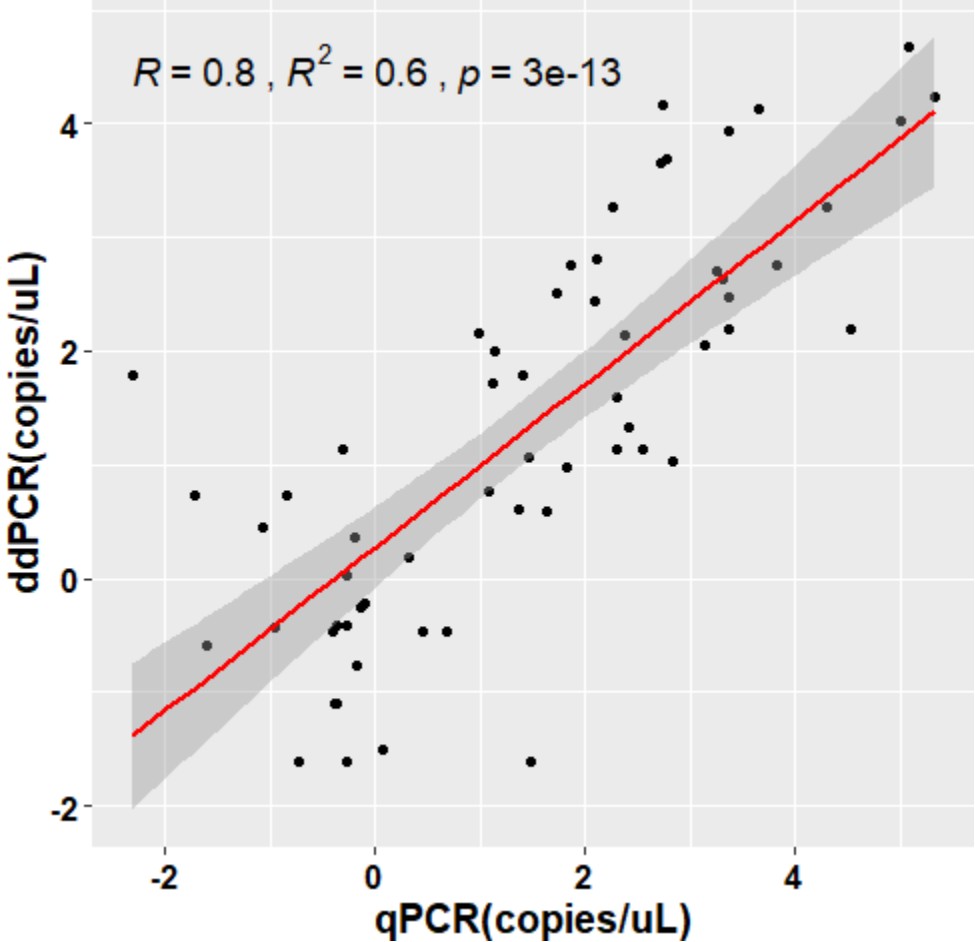

**FIG 4** Pearson correlation between qPCR and ddPCR assay in the detection of the copy numbers of the target DNA. Data is converted to ln(copies/uL).

multibacillary leprosy patients (33). Besides, numerous previous studies have reported higher sensitivity of the ddPCR assay in detecting infectious diseases, including malaria, tuberculosis, candidemia, and human schistosomiasis (34–39). The limit of detection of our ddPCR assay was found to be 0.1 parasite genome equivalents/reaction or ~11

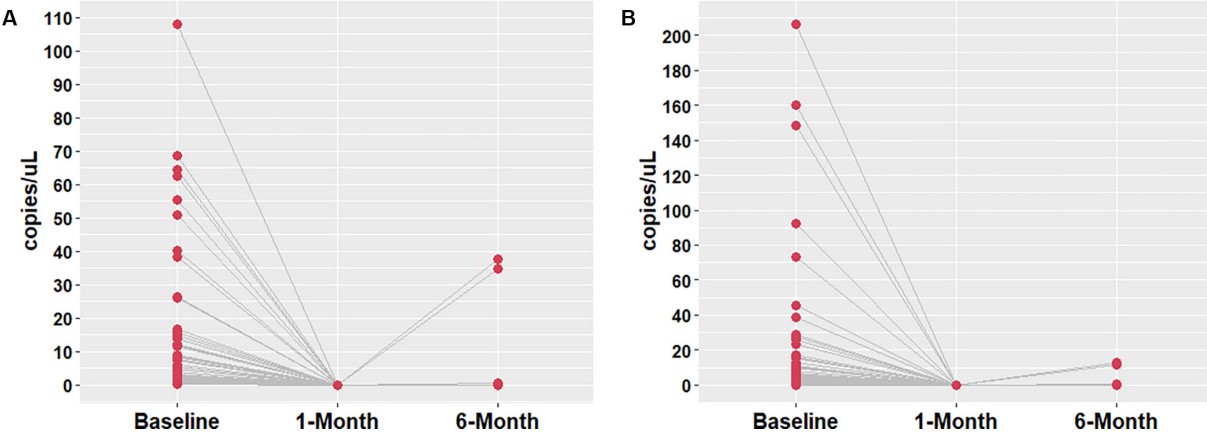

**FIG 5** (A) Disease monitoring with ddPCR assay. Detection of *Leishmania* DNA in three treated VL patients at 30 days and 180 days. (B) Disease monitoring with qPCR assay. Detection of *Leishmania* DNA in three treated VL patients at 30 days and 180 days.

parasites/mL, which is much lower than a previously observed limit of detection of 100 parasites/mL by ddPCR in the diagnosis of cutaneous leishmaniasis (40).

The CV or relative standard deviation generated from the inter-assay and intra-assay reproducibility test with the standard dilutions demonstrates excellent precision of the data, with the majority of dilutions <5% CV except for the highest dilution (Table 2). The inter-assay reproducibility test with six clinical samples also showed high precision, with a majority of samples yielding <10% CV. In general, <5% CV indicates high precision, whereas >20% CV indicates low precision (41). Several previous studies have demonstrated a higher reproducibility of ddPCR assays at lower CV values (42, 43). The higher CV value at the lowest concentration might have occurred because molecular assays tend to be unstable close to the limit of detection (44). Similar findings were demonstrated by Verheul et al. for a ddPCR assay detecting a muscular dystrophy target gene (45).

As expected, the qPCR assay yielded 100% sensitivity as observed in the previous studies (22, 46). To compare the parasite load between ddPCR and qPCR, we converted the parasite load of the qPCR assay to the absolute copy number of the target gene (18S rRNA gene) by multiplying the parasite load by 10, 50, and 100 to determine the trade-off copy number against which both the assays show the best agreement. According to a previous study, the target gene is present in 10 to 100 copies per Leishmania parasite (29). However, it would have been more specific to determine the number of target genes from the complete and annotated genome sequences of Indian *L. donovani* isolates.

In the reproducibility test, the $10^0$ dilutions, which contain one parasite genome equivalent/reaction, the ddPCR detected 8–20 copies with a median value of 12 copies. A Bland-Altman analysis indicated the best agreement at 10 copy numbers per parasite, with all values within a 2 SD limit and a mean difference (bias) of 0.06 between the observations (Fig. 2). This is lower than the accepted threshold of clinically significant variability ($0.5 \log_{10}$) (47). Moreover, the ICC value (0.8) was also higher than the other two conversions, which further demonstrates the best agreement. This is in accordance with previous studies, which also used the Bland-Altman analysis to compare pathogen detecting ddPCR and qPCR sensitivity (32, 42). In this study, we observed absolute concordance ($\kappa = 1.00$, McNemar's $P$ value = 1.00) between ddPCR and qPCR assays. Similarly, excellent concordance ($\kappa = 0.953$) was demonstrated between ddPCR and qPCR detecting SARS-CoV-2 RNA in plasma samples by Tedim et al. (48), who also reported a high correlation ($R^2 = 0.602$, $P < 0.0001$) between the copy numbers detected. A similar correlation was observed by Zhao et al. in the quantification of *Xanthomonas citri* subsp. *citri* (49).

Quantification of the parasite is imperative for treatment monitoring of the leishmaniasis patients. To date, ample direct and indirect biomarkers have been investigated to monitor the treatment outcome. In our previous study, we showed that the anti-rK39 antibody has the potential to predict the development of VL relapse and PKDL among treated VL patients (50). A systematic review by Kip et al. reported several antigen and antibody-based biomarkers for treatment monitoring (51). However, parasite DNA has been found to be the most accurate biomarker for monitoring the treatment outcome (51). To date, a number of clinical trials have utilized parasite DNA or qPCR for determining the parasite load in the blood of the VL patients as a pharmacodynamic assessment tool (52). In this study, for the first time, we evaluated ddPCR as a treatment monitoring tool for VL patients, following administration of AmBisome. Due to the variable efficacy of the existing anti-leishmanials, efforts are ongoing to develop new drugs (53, 54). As a proper treatment monitoring tool, the ddPCR could expedite the drug development pathway for VL and other forms of leishmaniasis.

On the Indian sub-continent, a single dose of Ambisome is recommended for treating primary VL cases (55). Despite the excellent efficacy of the regimen, almost 10% of the treated VL cases relapse within 6 months after the treatment (8). In addition, 5%–10% of cured VL patients develop PKDL within 2–4 years, which then acts as the potential reservoir for inter-epidemic disease transmission (56). So, monitoring the prognosis of

the diseases is as important as the early and accurate detection of cases to halt the transmission and eradicate the disease. We followed up on treated VL cases by both molecular assays at 1-mtp and 6-mtp. After 1-mtp, all of the patients were clinically cured, and none of the cases were found to be positive with either of the molecular assays. Our previous study also demonstrated circulatory parasite clearance in cured VL cases following effective treatment using qPCR as a diagnostic means (22). At 6-mtp, three cases were found to be positive in both qPCR and ddPCR assays, although all the patients remained clinically cured.

Since all of the VL patients were treated with an anti-leishmanial drug, it is unclear whether the detected DNA was from dead or viable parasites. In this regard, a threshold parasite load might give an indication toward successful treatment or chances for relapse in the future. However, it is difficult to determine such a trade-off for parasite abundance. In this study, the relapsed patients were detected with very low amounts of DNA. Similarly, in our previous studies, we found very low amounts of parasite DNA in the skin samples of PKDL patients (56, 57). In relapse cases, full parasite clearance is not achieved, where the remnant live parasites remain in a cryptic state in the leukocytes located in the spleen or bone marrow and at very low levels in the circulatory system (58). Such instances clearly indicate that a very low amount of live parasite can re-establish infection even after successful treatment. Therefore, the treated VL patients who are positive through real-time PCR/ddPCR at 6 months should be under close monitoring or follow-up for further interventions, if required.

Several factors can facilitate the proliferation of the pathogens and the re-emergence of the disease after successful treatment, such as ineffective cellular immunity after treatment due to immunosuppressive conditions, HIV, TB, malnutrition, inadequate treatment, and/or resistance to drugs (59). The pathogen severely alters the antigen processing and presentation capacities of dendritic cells and macrophages and escapes immune surveillance using an array of strategies yet to be fully understood (60). Several cytokines and immunoregulatory molecules could be potentially responsible for the persistence of the LD parasite and progression of the disease (58). *L. donovani* antigen-specific IgG1 levels were found to be significantly higher after 6 months of treatment in relapse VL than in cured cases in India and Sudan (61). A separate study conducted on VL patients in Bihar, India, demonstrated that IL-10 significantly correlates with high parasite burden and disease severity (62). Additionally, a study in a murine model reported that IL-4 and IL-10 induced a TH2 response that contributes to the logarithmic parasitic growth (63). In Brazil, VL relapsed patients have maintained lower CD4$^+$ T cells and higher IgG3 levels up to 6 months of the treatment compared to the non-relapsed VL (64).

ddPCR has been shown to have promising advantages for monitoring disease progression in several past studies (65–67). Roy et al. illustrated the potential of ddPCR technology in monitoring disease progression in *L. donovani* infection in PKDL patients (68). ddPCR was also found to be highly accurate and sensitive for measuring total HIV DNA and 2-LTR circles after antiretroviral therapy (69, 70). The reliability of ddPCR was shown for monitoring minimal residual disease in acute promyelocytic leukemia (71) and for predicting Philadelphia-positive acute lymphoblastic leukemia patients at risk of disease progression (72). Several other studies have also demonstrated ddPCR as an efficient disease monitoring tool upon treatment response (19, 73, 74).

Notwithstanding the excellent efficiency of ddPCR in treatment monitoring, currently, the cost and run time of this ultrasensitive assay are higher than that of qPCR (75). The estimated costs for in-house use of ddPCR reagents range from 5$ to 20$ per reaction, which appears to be above the range of in-house real-time PCR ($2/reaction) (35, 37). However, the capability of direct quantification of this method could obviate the development and purchase of costly reference material, which could lessen testing costs. Furthermore, its resistance to inhibitors could negate the need for repetitive runs for false negative samples, which might further decrease the time and the overall cost (17, 43, 76). Moreover, researchers are developing a multiplex ddPCR assay that could detect multiple targets in a small volume of sample, which can reduce the diagnostic costs along with

augmenting the sensitivity and making the diagnosis time-effective. In an earlier study, a multiplex ddPCR assay was proven to be three times cheaper than simplex qPCR and saved 6 working days in the time to result for the quantification of twelve approved GM maize lines (77). Full automation of the ddPCR method can further simplify the workflow and reduce hands-on time, which will be useful to achieve maximum implementation into routine clinical usage.

This study found ddPCR to be comparable to qPCR, which may pave the way to implement this ultrasensitive tool for the diagnosis of VL patients, treatment monitoring, and the surveillance of residual infection in the post-elimination era. The assay is a suitable surveillance tool to trace the transmission or reemergence of the disease in hotspots and identify potential new endemic areas. Moreover, this quantitative assay would be useful to gauge the *leishmania* infection in sandflies as a means of vector surveillance. However, further studies are warranted to ascertain the suitability of this ultrasensitive tool prior to its large-scale application.

## MATERIALS AND METHODS

### Study design, sites, and participants

The study was a laboratory-based diagnostic evaluation study with a case-control design, conducted using archived samples stored at −80°C and available from previous studies performed at icddr,b. Study participants' inclusion and sample collection were conducted during their corresponding studies at Surja Kanta Kala-azar Research Center situated in Mymensingh district, a highly endemic zone for VL that accounts for more than half of the total VL cases in Bangladesh. Laboratory activities were carried out at the Emerging Infections and Parasitology Laboratory, icddr,b, from January 2022 to December 2023. In total, 60 samples from confirmed VL cases and 60 samples from EHC were evaluated to determine the diagnostic efficiency of ddPCR, with the VL cases defined as individuals from endemic areas with less than 2 weeks of splenomegaly and a positive rK39 RDT test. In addition, individuals of either sex from the VL endemic area, with no history of VL/PKDL, clinically healthy without any symptoms of severe acute or chronic illness, including VL and PKDL, and rK39 RDT negative, were considered as EHC. The VL patients included in this study had been treated with liposomal amphotericin B, following the national guidelines (78), and all of them were followed up to 1 year post-treatment. Blood samples collected and archived from the same patients were subjected to investigation through ddPCR. In the previous studies, blood samples were collected from the patients before treatment (baseline) and after treatment at two time points: 30 days or 1-mtp, and 180 days or 6-mtp.

To determine the diagnostic efficiency of the investigative methods, the national guideline for VL diagnosis, along with effective treatment response, was considered as the gold standard in this study (78). Patients who developed a recurrence of the symptoms after 6 months but within 1 year of treatment were considered relapse cases. The DNA samples corresponding to the confirmed VL patients were used to assess the sensitivity, while DNA samples from healthy individuals were used to determine the specificity of ddPCR and qPCR assays. To assess the prognostic value of both assays, DNA isolates from blood samples of treated VL cases at 1-mtp and 6-mtp were used to detect Leishmania parasites by qPCR and ddPCR. Archived DNA samples were used to perform the molecular assays. In the previous studies, the DNA was extracted from 200 µL heparin-treated whole blood by DNeasy blood and tissue DNA extraction kits following the manufacturer's instructions (QIAGEN, Hilden, Germany) with minor modifications. Extracted DNA was eluted in 200 µL of elution buffer provided with the kits and stored at −80°C for prolonged storage in the Emerging Infections and Parasitology Laboratory, icddr,b.

## Real-time PCR for detection of LD DNA

The real-time PCR assay was performed following a method originally described by Vallur et al. (79). This Taqman PCR targets a conserved region of Leishmania REPL repeats (L42486.1) specific for *L. donovani* and *Leishmania infantum* coding for the 18S rRNA gene. A 20 µL reaction mix was prepared containing 9 µL template, 10 µL of TaqMan Gene Expression Master Mix (2×), and 1 µL pre-ordered primer-probe mix. Amplification was performed on a Bio-Rad CFX96 icycler system with the following reaction conditions: 10 min at 95℃, followed by 45 cycles of 15 s at 95℃ and 1 min at 60℃. To quantify the parasite load of each sample, each run included one standard curve with DNA concentrations ranging from 10,000 to 0.1 parasite genome equivalents/reaction. Each run also included one reaction with molecular-grade water as a negative control. Each DNA sample was analyzed in duplicate, and when indeterminate results were returned, one additional analysis was performed. Samples with a cycle threshold (Ct) >40 were considered negative.

## Droplet digital PCR for detection of LD DNA

The ddPCR assay was conducted on the QX200. For each assay, a final volume of 20 µL was prepared. This volume consisted of 10 µL of 2× ddPCR Supermix for Probes (no dUTP; Bio-Rad), 1 µL of a pre-ordered primer-probe mix (same as used for qPCR), and 9 µL of the extracted DNA. Each reaction, along with 70 µL of droplet generation oil, was loaded into an 8-well disposable cartridge (DG8). Subsequently, the cartridge was placed in the QX200 Droplet Generator (Bio-Rad) to generate droplets following the manufacturer's protocols. A droplet count of greater than 10,000 droplets was set as the cutoff when analyzing all ddPCR experiments (80). If the number of droplets was lower, the results were discarded, and the experiment was repeated. The generated droplets were then transferred to a 96-well plate for PCR amplification. Each run included a negative control reaction using molecular-grade water. The PCR consisted of 40 cycles, including a denaturation step at 95℃ for 5 min, followed by denaturation at 95℃ for 30 s, annealing at 60℃ for 1 min, and a final extension at 90℃ for 5 min. The amplification was performed on a Bio-Rad CFX96 iCycler system. Following amplification, the plate was then transferred to the QX200 to analyze the droplets, and subsequently, the data were processed using the QuantaSoft software. Thresholds between positive and negative droplets were adjusted manually based on clear differentiation of positive droplets from the negative cluster in individual wells, as well as using positive and no-template controls as a guide. The data were expressed as copies/µL and copies/reaction by the Quantasoft software.

## Analytical sensitivity, linearity, and reproducibility of droplet digital PCR assay

A 10-fold serial dilution series of DNA extracted from *in vitro* cultured *L. donovani* promastigotes was made from 1 ng to 1 fg of parasite DNA corresponding to 10,000 to 0.01 parasite genome equivalents/reaction to determine the minimal number of parasites that could be detected by the assay (22). Three replicates of DNA concentrations corresponding to 10,000 to 0.1 parasite genome equivalents/reaction ($10^4$–$10^{-1}$) were tested in a single run for intra-assay validation. For inter-assay validation, the same dilution series (10,000 to 0.1 parasite genome equivalents/reaction) and six clinical samples were run three times independently. Variability of the assay is measured and presented as the coefficient of variation (CV; shown as the percentage of the ratio of mean to standard deviation [SD]).

## Data analysis

All ddPCR data were generated using BioRad's QuantaSoft software version 1.7.4.0197. The clinical sensitivity and specificity of the real-time PCR and ddPCR assay for detecting and identifying Leishmania parasites were calculated considering clinical diagnosis

with effective treatment response as the "gold standard" for VL patients. Sensitivity and specificity with 95% CI were calculated using exact binomial methods for proportions. Cohen's kappa coefficient (k) was determined to test the agreement among the real-time PCR and ddPCR assay. According to the classification proposed by Landis and Koch, the values of Cohen's kappa coefficients were categorized as excellent (1.00–0.81); good (0.80–0.61); moderate (0.60–0.41); weak (0.40–0.21); and negligible agreement (0.20–0.00) (81). The McNemar test was performed to evaluate the concordance and discordance between the exploratory methods. Bland-Altman analysis was performed to compare the parasite loads determined through ddPCR and real-time PCR assay. According to the literature, each *L. donovani* genome contains 10 to 200 copies of the ribosomal 18S rRNA gene (82, 83). In qPCR, the number of parasites is quantified against the parasite DNA standard, whereas the ddPCR quantifies the absolute copy numbers of the target DNA. To convert the qPCR results to the absolute number of targets, the qPCR values were multiplied by 10, 50, and 100 copies of the 18S rRNA gene per Leishmania genome, respectively. Besides, the dynamics of parasite burden in each of the treated VL patients were determined through both the molecular methods. Parametric or nonparametric statistics were performed with the quantitative variables based on the distribution of data. A *P* value <0.05 was considered statistically significant. Data analysis was performed using R version software, GraphPad Prism, and IBM SPSS Statistics version 22.0.

## ACKNOWLEDGMENTS

We are grateful to all of the participants for their valuable participation in this study. The authors are thankful to core donors, including the Government of the People's Republic of Bangladesh; Global Affairs Canada (GAC), Canada; Swedish International Development Cooperation Agency (Sida); and the Department for International Development (UKAid) for providing unrestricted support and commitment to icddr,b's research efforts.

The funding was granted by the Swedish International Development Cooperation Agency (Sida, GR-01455), Sweden, and Rainy Day Grant Fund under "Young investigator's award," the International Centre for Diarrhoeal Disease Research, Bangladesh (icddr,b) to R.C. The authors alone are responsible for the views expressed in this manuscript. The funders had no role in study design, data collection and analysis, decision to publish, or preparation of the manuscript.

P.G., D.M., R.C., and M.A.R. conceptualized the study. M.A.R. and R.C. visualized the study. A.S., F.T.G., M.S.C., F.H., N.T.M., M.K., and M.A.A.C. were responsible for the data curation. M.A.R., R.C., R.M.K., A.C., and P.B. analyzed the data. P.G., R.C., and D.M. were responsible for supervision and funding acquisition. A.S., F.T.G., M.S.C., F.H., N.T.M., M.K., P.B., and M.A.A.C. performed the methods. M.A.R., P.G., and R.C. wrote the original draft. A.A.E.W., M.W., D.M., and M.D. reviewed and edited the manuscript. All authors read and approved the final manuscript.

The authors have declared that no competing interests exist.

## AUTHOR AFFILIATIONS

[1]Emerging Infections and Parasitology Laboratory, Nutrition Research Division, International Centre for Diarrhoeal Disease Research Bangladesh (icddrb), Dhaka, Bangladesh

[2]Institute of Animal Hygiene and Veterinary Public Health, Leipzig University, Leipzig, Germany

[3]Department of Empirical Health Economics, Technische Universität Berlin, Berlin, Germany

[4]Department of Zoology, Faculty of Biological Sciences, University of Dhaka, Dhaka, Bangladesh

[5]HDT Bio, Seattle, Washington, USA

## AUTHOR ORCIDs

Prakash Ghosh ⓘ http://orcid.org/0000-0002-4300-8761
Ahmed Abd El Wahed ⓘ https://orcid.org/0000-0003-3347-6075
Dinesh Mondal ⓘ https://orcid.org/0000-0001-9154-8327

## AUTHOR CONTRIBUTIONS

Md. Abu Rahat, Conceptualization, Formal analysis, Visualization, Writing – original draft | Prakash Ghosh, Conceptualization, Funding acquisition, Supervision, Writing – original draft | Rajashree Chowdhury, Conceptualization, Formal analysis, Funding acquisition, Supervision, Visualization, Writing – original draft | Ashik Sharfaraz, Data curation, Methodology | Fariha Tasneem Gawhar, Data curation, Methodology | Md. Sakib Chowdhury, Data curation | Faria Hossain, Data curation, Methodology | Nishad Tasnim Mithila, Data curation, Methodology | Mostafa Kamal, Data curation, Methodology | Md. Arko Ayon Chowdhury, Data curation, Methodology | Priyanka Barua, Formal analysis, Methodology | Rea Maja Kobialka, Formal analysis | Arianna Ceruti, Formal analysis | Ahmed Abd El Wahed, Writing - review and editing | Manfred Weidmann, Writing - review and editing | Malcolm Duthie, Writing - review and editing | Dinesh Mondal, Conceptualization, Funding acquisition, Supervision, Writing - review and editing

## DATA AVAILABILITY

Most of the data rendering the study are included within the article. Further inquiries can be directed to the corresponding authors.

## ETHICS APPROVAL

This study was approved by the Institutional Review Board (IRB) and Ethical Review Committee (ERC) of the International Centre for Diarrhoeal Disease Research, Bangladesh (icddr,b), Dhaka, Bangladesh (PR-21120). Written informed consent was obtained from each individual and/or minor's legal guardians, conferring use of their archived clinical samples for VL research.

## ADDITIONAL FILES

The following material is available online.

Open Peer Review

**PEER REVIEW HISTORY (review-history.pdf).** An accounting of the reviewer comments and feedback.

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
