## [Reviewer comments · Microbiology Spectrum]

Microbiology Spectrum

Application of ddPCR for diagnosis and treatment monitoring of visceral leishmaniasis patients: addition of an ultrasensitive tool to the diagnostic arsenal

Md. Rahat, Prakash Ghosh, Rajashree Chowdhury, Ashik Sharfaraz, Fariha Tasneem Gawhar, Md. Sakib Chowdhury, Faria Hossain, Nishad Mithila, Mostafa Kamal, Md. Arko Chowdhury, Priyanka Barua, Rea Maja Kobiarka, Arianna Ceruti, Ahmed Abd El Wahed, Manfred Weidmann, Malcolm Duthie, and Dinesh Mondal

Corresponding Author(s): Prakash Ghosh, Universitat Leipzig

Review Timeline:

Submission Date:	June 13, 2025
Editorial Decision:	September 1, 2025
Revision Received:	October 30, 2025
Accepted:	December 2, 2025

Editor: William Witola

Reviewer(s): Disclosure of reviewer identity is with reference to reviewer comments included in decision letter(s). The following individuals involved in review of your submission have agreed to reveal their identity: Ariful Basher (Reviewer #1); Alejandro Schijman (Reviewer #2)

Transaction Report:

DOI: <https://doi.org/10.1128/spectrum.01800-25>

Re: Spectrum01800-25 (**Application of ddPCR for diagnosis and treatment monitoring of visceral leishmaniasis patients: addition of an ultrasensitive tool to the diagnostic arsenal**)

Dear Dr. Prakash Ghosh:

Thank you for the privilege of reviewing your work. Below you will find my comments, instructions from the Spectrum editorial office, and the reviewer comments.

Revision Guidelines

Sincerely,
William Witola
Editor
Microbiology Spectrum

Reviewer #1 (Comments for the Author):

The authors should declare the conflict of interest regarding the use of BIORAD. s kits

Introduction:

R 116 to overcome qPCR ; sentence better omitted as study findings do not reflect better, it may be in a future direction.

R 142,143 59 had splenomegaly, but in table 1 all had. Confused about the sentence less than two weeks , better to rewrite

R 145 reported relapse by how clinically or by DDPCR?

Discussion :

R 210 rk39 value decline already mentioned in the introduction with a different reference.

ddPCR is very promising, the authors mention the clinical utility, especially monitoring the patients, but I think it may be clearer that such a low level identification may not be confused with death parasite DNA and further study is needed to find out a treatment threshold.

Table 1 is not thought to be relevant to the study.

Table 2 is also irrelevant in my thought, as it is already mentioned in the text that the two are 100% sensitive and specific in comparison to each other and the serology test.

Reviewer #2 (Comments for the Author):

First study to report the use of ddPCR for diagnosis and treatment monitoring of visceral leishmaniasis patients.

The potential of this methodology is interesting. However, the high cost of the equipment may hamper its implementation which will be able to be carried out only in reference centers. Issues regarding cost-benefit should be discussed in the manuscript.

The written English should be revised. Some suggestions have been done using the track changes system and attached.

Materials and Methods:

Can you explain how is calculated the equivalence of parasite genome equivalents per reaction and fg of genomic DNA ?

Could you specify which guide has been used to estimate intra-assay and inter-assay CVs?

Could you clarify the rationale for estimating parasitic loads based on the assumption of 10, 50, or 100 copies of the target sequence per *Leishmania* genome? Why not instead use the validated gene dosage information available from sequenced *Leishmania* genomes and apply the copy number already reported in GenBank?

Caption to Figure 2 . Please clarify if the LOD exhibited in the standard curve is actually the limit of quantification... In general, LoD means LoD 95 and is calculated by performing a series of replicates along different working days, following the CLSI guides. The standard curve with six-dilution range (10,000 - 0.1 parasite/s per reaction) exhibiting the limit of detection (LOD) of ddPCR.

When you express sensitivity values lower than 1 parasite per reaction, it should be expressed as parasite genome equivalents/reaction.

Minor Revision

Line 165.. The CV values demonstrate a high reproducibility in all the but the samples with very low DNA copies.

Table 1

Please replace (n%) by n (%)

The authors should declare the conflict of interest regarding the use of BIORAD. s kits

Introduction:

R-116 to overcome qPCR ; sentence better omitted as study findings do not reflect better, it may be in a future direction.

R -142,143 59 had splenomegaly, but in table -1 all had. Confused about the sentence less than two weeks , better to rewrite

R-145 reported relapse by how clinically or by DDPCR?

Discussion :

R-210 rk39 value decline already mentioned in the introduction with a different reference.

ddPCR is very promising, the authors mention the clinical utility, especially monitoring the patients, but I think it may be clearer that such a low-level identification may not be confused with death parasite DNA and further study is needed to find out a treatment threshold.

Table 1 is not thought to be relevant to the study.

Table 2 is also irrelevant in my thought, as it is already mentioned in the text that the two are 100% sensitive and specific in comparison to each other and the serology test.

[revised manuscript text omitted]

that a multiplex ddPCR assay was 3 times cheaper than simplex qPCR and reduced the time to
results by six working days when quantifying twelve approved GM maize lines⁸³.

Full automation of the ddPCR method can further simplify the work-flow and can reduce hands-
on-time, which will be useful to achieve optimal implementation into routine clinical usage.

This study found ddPCR to be comparable to qPCR, which may pave the way to implement this
ultrasensitive tool for diagnosis of VL patients, treatment monitoring, and the surveillance of
residual infection in the post elimination era. The assay is a suitable surveillance tool to trace the
transmission or reemergence of the disease in hotspots and identify potential new endemic areas.
Moreover, this quantitative assay would be useful to gauge *leishmania* infection in sandflies as
means of vector surveillance. However, further studies are warranted to ascertain the suitability
of this ultrasensitive tool prior its large-scale application.

**Methods and Materials**

**Ethics statement**

This study was approved by the institutional Review Board (IRB) and Ethical Review
Committee (ERC) of International Centre for Diarrhoeal Disease Research, Bangladesh (icddr,b),
Dhaka, Bangladesh (PR-21120). Written informed consent was obtained from each individual
and/or minors legal guardians conferring use of their archived clinical samples for VL research.

**Study design, sites and participants**

The study was a laboratory-based diagnostic evaluation study with case-control design,
conducted using archived samples stored at -80°C and available from previous studies performed
at icddr, b. Study participants inclusion and sample collection were conducted during their
corresponding studies at Surja Kanta Kala-azar Research Centre (SKKRC) situated in
Mymensingh district, a highly endemic zone for VL that accounts for more than half of the total
VL cases in Bangladesh. Laboratory activities were carried out at the Emerging Infections and
Parasitology Laboratory, icddr,b from January 2022 to December 2023. In total, 60 samples from
confirmed VL cases and 60 samples from endemic healthy controls (EHC) were evaluated to
determine the diagnostic efficiency of ddPCR, with the VL cases defined as individuals from
endemic area with less than two weeks of splenomegaly and positive rK39 RDT test. In addition,
individuals of either sex from the VL endemic area, with no history of VL/PKDL, clinically
healthy without any symptoms of severe acute or chronic illness including VL and PKDL, and
rK39 RDT negative were considered as EHC. The VL patients included in this study had been
treated with liposomal amphotericin B following the national guidelines²² and all of them were
followed up to 1 year post treatment. Blood samples collected and archived from the same
patients were subjected to investigation through ddPCR. In the previous studies, blood samples
were collected from the patients before treatment (baseline) and after treatment at two time
points: 30 day or one month time point (1-mtp), and 180 day or six month time point (6-mtp).

To determine the diagnostic efficiency of the investigative methods, the national Guideline for
VL diagnosis along with effective treatment response was considered as the gold standard in this
study²². Patients who developed reoccurrence of the symptoms after 6 months but within the 1
366 year of treatment were considered as relapsed cases. The DNA samples corresponding to the
367 confirmed VL patients were used to assess the sensitivity, while DNA samples from healthy
individuals were used to determine the specificity of ddPCR and qPCR assays. To assess the
prognostic value of both assays, DNA isolates from blood samples of treated VL cases at 1-mtp
and 6-mtp were used to detect *Leishmania* parasites by qPCR and ddPCR. Archived DNA
samples were used to perform the molecular assays. In the previous studies, the DNA was

extracted from 200 μ L heparin-treated whole blood by DNeasy blood and tissue DNA extraction
kits following the manufacturer's instructions (QIAGEN, Hilden, Germany) with minor
modifications. Extracted DNA was eluted in 200 μ L of elution buffer provided with the kits and
stored at -80°C for prolonged storage in Emerging Infections and Parasitology Laboratory,
icddr,b.

**Real-time PCR for detection of LD DNA:**

The real time PCR assay was performed following a method originally described by Vallur et al,
2014²³. This Taqman PCR targets a conserved region of Leishmania REPL repeats (L42486.1)
specific for *L. donovani* and *L. infantum* coding for the 18S rRNA gene. A 20 μ L reaction mix
was prepared containing 9 μ L template, 10 μ L of TaqMan® Gene Expression Master Mix (2X),
and 1 μ L pre-ordered primer-probe mix. Amplification was performed on a Bio-rad CFX96
icycler system under the following reaction conditions: 10 min at 95°C , followed by 45 cycles
of 15 seconds at 95°C and 1 min at 60°C . To quantify the parasite load of each sample, each run
included one standard curve with DNA concentrations ranging from 10000 to 0.1 parasites per
reaction. Each run also included one reaction with molecular grade water as a non-template
control. Each DNA sample was analyzed in duplicate and when indeterminate results were
obtained, an additional analysis was performed. Samples with cycle threshold (Ct) >40 were
considered as negative.

**Droplet digital PCR for detection of LD DNA:**

The ddPCR assay was conducted on the QX200. For each assay, a final volume of 20 μ L was
prepared. This volume consisted of 10 μ L of 2x ddPCR Supermix for Probes (no dUTP; Bio-
Rad), 1 μ L of a pre-ordered primer-probe mix (same as used for qPCR), and 9 μ L of the
extracted DNA. Each reaction, along with 70 μ L of droplet generation oil, was loaded into an 8-
well disposable cartridge (DG8). Subsequently the cartridge was placed in the QX200 Droplet
Generator (Bio-Rad) to generate droplets following the manufacturer's protocols. A droplet count
of greater than 10,000 droplets was set as the cutoff when analyzing all ddPCR experiments²⁴. If
the number of droplets was lower, the results were discarded and the experiment was repeated.
The generated droplets were then transferred to a 96-well plate for PCR amplification. Each run
included a negative control reaction using molecular grade water. The PCR consisted of 40
cycles, including a denaturation step at 95°C for 5 minutes, followed by denaturation at 95°C for

30 seconds, annealing at 60°C for 1 minute, and a final extension at 90°C for 5 minutes. The
amplification was performed on a Bio-Rad CFX96 iCycler system. Following amplification, the
plate was then transferred to the QX200 to analyze the droplets, and subsequently the data were
processed using the QuantaSoft software. Thresholds between positive and negative droplets
were adjusted manually based on clear differentiation of positive droplets from the negative
cluster in individual wells as well as using positive and no-template controls as a guide. The data
was expressed as copies/ μ L and copies/reaction by the Quantasoft software.

**Analytical Sensitivity, linearity and reproducibility of droplet digital PCR assay:**

A 10-fold serial dilution series of DNA extracted from in vitro cultured *L. donovani*
promastigotes was made from 1 ng to 1fg of parasite DNA corresponding to 10,000 to 0.01
parasites per reaction to determine the minimal number of parasites that could be detected by the
assay. Three replicates of DNA concentrations corresponding to 10,000 to 0.1 parasites per
reaction (10^4 - 10^{-1}) were tested in a single run for intra-assay validation. For inter-assay
validation, the same dilution series (10,000 to 0.1 parasites per reaction) and six clinical samples
were run three times independently. Variability of the assay is measured and presented as
coefficient of variation (CV; shown as the percentage of the ratio of mean to standard deviation
(SD)).

**Data Analysis:**

All ddPCR data were generated using BioRad's QuantaSoft™ software version 1.7.4.0197. The
clinical sensitivity and specificity of the real time PCR and ddPCR assay for detecting and
identifying Leishmania parasites were calculated considering clinical diagnosis with effective
treatment response as the “gold standard” for VL patients. Sensitivity and specificity with 95%
CI were calculated using exact binomial methods for proportions. Cohen's kappa coefficient (k)
was determined to test the agreement among real time PCR and ddPCR assay. According to the
classification proposed by Landis and Koch, the values of Cohen's kappa coefficients were
categorized as excellent (1.00–0.81); good (0.80–0.61); moderate (0.60–0.41); weak (0.40–0.21);
and negligible agreement (0.20–0.00)²⁵. The McNemar test was performed to evaluate the
concordance and discordance between the exploratory methods. Bland–Altman Analysis was
performed to compare the parasite loads determined through ddPCR and real-time PCR assay.
According to the literature, each *L. donovani* genome contains 10 to 200 copies of the ribosomal

18SrRNA gene^{26,27}. In qPCR, the number of parasite genome equivalents is quantified against
the parasite DNA standard, whereas the ddPCR quantifies the absolute copy numbers of the
target DNA. To convert the qPCR results to the absolute number of targets, the qPCR values
were multiplied by 10, 50 and 100 copies of the 18SrRNA gene per Leishmania genome,
respectively. Besides, the dynamics of parasite burden in each of the treated VL patients was
determined through both of the molecular methods. Parametric or nonparametric statistics were
performed with the quantitative variables based on the distribution of data. A p-value < 0.05 was
considered as statistically significant. Data analysis was performed using R version software,
Graphpad Prism and IBM SPSS Statistics version 22.0.

**References**

- 1. Sasidharan, S. & Saudagar, P. Leishmaniasis: where are we and where are we heading?
*Parasitol. Res.* **120**, 1541–1554 (2021).
- 2. Maruf, S. *et al.* Revisiting the diagnosis and treatment of Para Kala-azar Dermal
Leishmaniasis in the endemic foci of Bangladesh. *PLoS One* **18**, 1–9 (2023).
- 3. Wamai, R. G., Kahn, J., McGloin, J. & Ziaggi, G. Visceral leishmaniasis: a global
overview. *J. Glob. Heal. Sci.* **2**, 1–22 (2020).
- 4. Ahmed, B.-N. *et al.* Kala-azar (visceral leishmaniasis) elimination in Bangladesh:
successes and challenges. *Curr. Trop. Med. Reports* **1**, 163–169 (2014).
- 5. World Health Organization. *Bangladesh achieves historic milestone by eliminating kala-*
*azar as a public health problem.* [https://www.who.int/news/item/31-10-2023-bangladesh-](https://www.who.int/news/item/31-10-2023-bangladesh-achieves-historic-milestone-by-eliminating-kala-azar-as-a-public-health-problem?)
[achieves-historic-milestone-by-eliminating-kala-azar-as-a-public-health-problem?](https://www.who.int/news/item/31-10-2023-bangladesh-achieves-historic-milestone-by-eliminating-kala-azar-as-a-public-health-problem?) (2023).
- 6. Courtenay, O., Peters, N. C., Rogers, M. E. & Bern, C. Combining epidemiology with
basic biology of sand flies, parasites, and hosts to inform leishmaniasis transmission
dynamics and control. *PLoS Pathog.* **13**, (2017).
- 7. Bhattacharya, S. K. & Dash, A. P. Elimination of kala-azar from the Southeast Asia
Region. *Am. J. Trop. Med. Hyg.* **96**, 802–804 (2017).

- 8. Burza, S. *et al.* Risk factors for visceral leishmaniasis relapse in immunocompetent
patients following treatment with 20 mg/kg liposomal amphotericin B (Ambisome) in
Bihar, India. *PLoS Negl. Trop. Dis.* **8**, e2536 (2014).
- 9. Cloots, K. *et al.* Impact of the visceral leishmaniasis elimination initiative on *Leishmania*
*donovani* transmission in Nepal: a 10-year repeat survey. *Lancet Glob. Heal.* **8**, e237–
e243 (2020).
- 10. Saurabh, S. Time for a village-level strategy for the elimination of kala-azar (visceral
leishmaniasis) in India: analysis of potential kala-azar outbreak situation in 2018. *Trop.*
*Doct.* **51**, 84–91 (2021).
- 11. Kumar, A., Saurabh, S., Jamil, S. & Kumar, V. Intensely clustered outbreak of visceral
leishmaniasis (kala-azar) in a setting of seasonal migration in a village of Bihar, India.
*BMC Infect. Dis.* **20**, 1–13 (2020).
- 12. Cloots, K. *et al.* Diagnosis of visceral leishmaniasis in an elimination setting: A validation
study of the diagnostic algorithm in India. *Diagnostics* **12**, 670 (2022).
- 13. Kassa, M. *et al.* Diagnostic accuracy of direct agglutination test, rk39 elisa and six rapid
diagnostic tests among visceral leishmaniasis patients with and without hiv coinfection in
ethiopia. *PLoS Negl. Trop. Dis.* **14**, 1–13 (2020).
- 14. Hindson, B. J. *et al.* High-throughput droplet digital PCR system for absolute quantitation
of DNA copy number. *Anal. Chem.* **83**, 8604–8610 (2011).
- 15. Rutsaert, S., Bosman, K., Trypsteen, W., Nijhuis, M. & Vandekerckhove, L. Digital PCR
as a tool to measure HIV persistence. *Retrovirology* **15**, 1–8 (2018).
- 16. Miotke, L., Lau, B. T., Rumma, R. T. & Ji, H. P. High sensitivity detection and
quantitation of DNA copy number and single nucleotide variants with single color droplet
digital PCR. *Anal. Chem.* **86**, 2618–2624 (2014).
- 17. Dingle, T. C., Sedlak, R. H., Cook, L. & Jerome, K. R. Tolerance of droplet-digital PCR
vs real-time quantitative PCR to inhibitory substances. *Clin. Chem.* **59**, 1670–1672

- (2013).
- 18. Sedlak, R. H., Kuypers, J. & Jerome, K. R. A multiplexed droplet digital PCR assay
performs better than qPCR on inhibition prone samples. *Diagn. Microbiol. Infect. Dis.* **80**,
285–286 (2014).
- 19. Zheng, W. W. C., Liu, Z., Zhao, C. B., Xu, G. L. Z. & Wang, Z. J. C. H. Droplet digital
PCR for BCR / ABL (P210) detection of chronic myeloid leukemia : A high sensitive
method of the minimal residual disease and disease progression. 291–296 (2018)
doi:10.1111/ejh.13084.
- 20. Maheshwari, Y., Selvaraj, V., Hajeri, S. & Yokomi, R. Application of droplet digital PCR
for quantitative detection of *Spiroplasma citri* in comparison with real time PCR. *PLoS*
*One* **12**, e0184751 (2017).
- 21. Ma, J., Li, N., Guarnera, M. & Jiang, F. Quantification of plasma miRNAs by digital PCR
for cancer diagnosis. *Biomark. Insights* **8**, 127–136 (2013).
- 22. Welfare, F. National Guideline For Kala-azar Case Management National Guideline For
Kala-azar Case Management. (2013).
- 23. Vallur, A. C. *et al.* Biomarkers for intracellular pathogens: establishing tools as vaccine
and therapeutic endpoints for visceral leishmaniasis. *Clin. Microbiol. Infect.* **20**, O374–
O383 (2014).
- 24. Pinheiro, L. B. *et al.* Evaluation of a droplet digital polymerase chain reaction format for
DNA copy number quantification. *Anal. Chem.* **84**, 1003–1011 (2012).
- 25. Ghosh, P. *et al.* Evaluation of diagnostic performance of rK28 ELISA using urine for
diagnosis of visceral leishmaniasis. *Parasit. Vectors* **9**, 1–8 (2016).
- 26. Pasquier, G., Andreotti, Q., Ravel, C. & Sterkers, Y. Molecular diagnosis of visceral
leishmaniasis: a French study comparing a reference PCR method targeting kinetoplast
DNA and a commercial kit targeting ribosomal DNA. *Microbiol. Spectr.* **11**, e02154-23
(2023).

- 27. Yan, S., Lodes, M. J., Fox, M., Myler, P. J. & Stuart, K. Characterization of the
*Leishmania donovani* ribosomal RNA promoter. *Mol. Biochem. Parasitol.* **103**, 197–210
(1999).
- 28. WHO. Bangladesh achieves historic milestone by eliminating kala-azar as a public health
problem. in (2023).
- 29. Shetty, D. *Bangladesh Becomes World's First Country to Eliminate Visceral*
*Leishmaniasis*. [https://healthpolicy-watch.news/bangladesh-becomes-worlds-first-country-](https://healthpolicy-watch.news/bangladesh-becomes-worlds-first-country-to-eliminate-visceral-leishmaniasis/)
[to-eliminate-visceral-leishmaniasis/](https://healthpolicy-watch.news/bangladesh-becomes-worlds-first-country-to-eliminate-visceral-leishmaniasis/).
- 30. Parikh, R., Mathai, A., Parikh, S., Sekhar, G. C. & Thomas, R. Understanding and using
sensitivity, specificity and predictive values. *Indian J. Ophthalmol.* **56**, 45–50 (2008).

[revised manuscript text omitted]

662

663 Table 02: Sensitivity and specificity of qPCR and ddPCR

664

Methods	Sensitivity (%) (n/N)	Specificity (%) (n/N)
ddPCR	100 (94.04-100.00) (60/60)	100(94.04-100.00) (60/60)
qPCR	100 (94.04-100.00) (60/60)	100 (94.04-100.00) (60/60)

665

666 Table 03: Repeatability and reproducibility of the ddPCR assay

667

Parasite load	Inter assay variation of copies per reaction						Intra assay variation of copies per reaction					
	Assay 1	Assay 2	Assay 3	Mean	SD	CV	Replicate 1	Replicate 2	Replicate 3	Mean	SD	CV
1x10 ⁴	172000	194000	166400	177466.67	14589.49	8.22	174000	149800	194000	172600	22133.23	12.82
1x10 ³	17620	16280	17880	17260	858.6	4.97	17160	17840	16780	17260	537.03	3.11
1x10 ²	1640	1686	1742	1689.33	51.08	3.02	1640	1680	1620	1646.67	30.55	1.86
1x10 ¹	142	154	150	148.67	6.11	4.11	194	230	204	209.33	18.58	8.88
1x10 ⁰	12	13.2	8.8	11.33	2.27	20.07	20	20	18	19.33	1.15	5.97
1x10 ⁻¹	0	3.6	1.2	1.6	1.83	114.56	0	2	2	1.33	1.15	86.6
ID-01	76	100	90	88.67	12.06	13.6	NA	NA	NA	NA	NA	NA

ID-02	88	94	94	92	3.46	3.77	NA	NA	NA	NA	NA	NA
ID-03	24.6	16.4	24	21.67	4.57	21.1	NA	NA	NA	NA	NA	NA
ID-04	11.4	9.4	6.6	9.13	2.41	26.4	NA	NA	NA	NA	NA	NA
ID-05	6.6	6.2	14	8.93	4.39	49.17	NA	NA	NA	NA	NA	NA
ID-06	866	812	864	847.33	30.62	3.61	NA	NA	NA	NA	NA	NA

Figures:

A

B

Figure 01: Results from ddPCR experiments. A) Representative bar graph of ddPCR analysis
from blood DNA samples for identifying *Leishmania* DNA. Light green bars represent total
droplets number and blue bars represent the positive droplets. B) One-dimensional scatter plots
showing amplitude of fluorescence on the Y-axis and event number on the X-axis for the
positive droplets.

Figure 02: The standard curve with six-dilution range (10,000 - 0.1 parasite/s per reaction)
exhibiting the limit of detection (LOD) of ddPCR.

Figure 03-A: Bland-Altman analysis to determine the agreement between qPCR and ddPCR
assay. (unit= copies/uL). Assuming 10 copies present in one parasite and converting the value to
$\ln(\text{copies/uL})$.

Figure 03-B: Bland-Altman analysis to determine the agreement between qPCR and ddPCR assay. (unit= copies/uL). Assuming 50 copies present in one parasite and converting the value to $\ln(\text{copies/uL})$.

Figure 03-C: Bland-Altman analysis to determine the agreement between qPCR and ddPCR assay. (unit= copies/uL). Assuming 100 copies present in one parasite and converting the value to $\ln(\text{copies/uL})$.

Figure 04: Pearson correlation between qPCR and ddPCR assay in detection of the copy numbers of the target DNA. Data is converted to $\ln(\text{copies/uL})$.

Figure 05-A: Disease monitoring with ddPCR assay. Detection of leishmania DNA in three
treated VL patients at 30 day and 180 day.

Figure 05-B: Disease monitoring with qPCR assay. Detection of leishmania DNA in three treated
VL patients at 30 day and 180 day.

Acknowledgements

We are grateful to all of the participants for their valuable participation in this study. The authors are thankful to core donors include the Government of the People’s Republic of Bangladesh; Global Affairs Canada (GAC), Canada; Swedish International Development Cooperation Agency (Sida); and the Department for International Development (UKAid) for providing unrestricted support and commitment to icddr,b’s research efforts.

Authors contributions

PG, DM, RC and MAR conceptualised the study. MAR and RC visualisation. AS, FTG, MSC, FH, NTM, MK and MAAC were responsible for the data curation. MAR, RC, RMK and AC, PB analysed the data. PG, RC and DM were responsible for supervision and funding acquisition. AS, FTG, MSC, FH, NTM, MK, PB and MAAC performed the methods. MAR, PG and RC wrote the original draft. AAEW, MW, DM and MD reviewed and edited the manuscript. All authors read and approved the final manuscript.

Data availability statement

Most of the data rendering the study are included within the article. Further inquiries can be directed to the corresponding authors.

Disclosure statement

The authors have declared that no competing interests exist.

Funding

The funding was granted by the Swedish International Development Cooperation Agency(Sida,GR-01455), Sweden and Rainy Day Grant Fund under ‘Young investigator’s award’, International Centre for Diarrhoeal Disease Research, Bangladesh (icddr,b) to RC. The authors alone are responsible for the views expressed in this manuscript. The funders had no role in study design, data collection and analysis, decision to publish, or preparation of the manuscript.

Reviewer #1 (Comments for the Author):

Comment: The authors should declare the conflict of interest regarding the use of BIORADs kits.

Response: The kits were purchased from BIORAD to perform the ddPCR assay. The company didn't have any contribution in the designing the study, developing the assay and data analysis. Therefore, the authors think that it is not necessary to declare any conflict of interest regarding the use of BIORADs kits.

Comment: R 116 to overcome qPCR; sentence better omitted as study findings do not reflect better, it may be in a future direction.

Response: The line is revised accordingly in the manuscript.

The sentence reflects the aim of the study in principle. The ddPCR assay has outperformed the conventional real-time PCR assay in different scenarios. In the current study, we observed 100% agreement between ddPCR and Real-time PCR which has been discussed in the discussion section. A minor change has been made in the sentence.

Comment: R 142,143, 59 had splenomegaly, but in table 1 all had. Confused about the sentence less than two weeks, better to rewrite.

Response: Correction has been made in the table.

Comment: R 145 reported relapse by how clinically or by DDPCR?

Response: Relapse was detected based on the clinical features along with the parasitological confirmation made by real-time PCR.

Comment: R 210 rk39 value decline already mentioned in the introduction with a different reference.

Response: Thanks for your observation. The same reference has been provided.

Comment: ddPCR is very promising, the authors mention the clinical utility, especially monitoring the patients, but I think it may be clearer that such a low level identification may not be confused with death parasite DNA and further study is needed to find out a treatment the threshold.

Response: You are absolutely correct. Really appreciate your input here.

Since all of the VL patients were treated with anti-leishmanial drug, it is unclear whether the detected DNA were from dead or viable parasites. Regarding the threshold, it is difficult to determine such trade-off a parasite abundance. Here to note, the relapsed patients were detected with very low amount of DNA in this study. Besides, in our previous studies we found very low amount of parasite DNA in the skin samples of PKDL patients. Such instances clearly indicate that a very low amount of live parasite can re-establish infection

even after successful treatment. Therefore, the treated VL patients being positive through real-time/ddPCR at 6 months should be under close monitoring or follow-up for further interventions, if required.

It has been included in the discussion section.

Comment: Table 1 is not thought to be relevant to the study.

Response: Since the study findings have significant clinically implications, the authors would like to keep the table.

Comment: Table 2 is also irrelevant in my thought, as it is already mentioned in the text that the two are 100% sensitive and specific in comparison to each other and the serology test.

Response: The table 2 is omitted.

Reviewer #2 (Comments for the Author):

Comment: First study to report the use of ddPCR for diagnosis and treatment monitoring of visceral leishmaniasis patients.

Response: We appreciate your observation.

Comment: The potential of this methodology is interesting. However, the high cost of the equipment may hamper its implementation which will be able to be carried out only in reference centers. Issues regarding cost-benefit should be discussed in the manuscript.

Response: Thank you for your comment. The mentioned perspective has already been discussed in the discussion section (line number: 311-324).

Comment: The written English should be revised. Some suggestions have been done using the track changes system and attached.

Response: The necessary corrections have been made in the revised version.

Materials and Methods:

Comment: Can you explain how is calculated the equivalence of parasite genome equivalents per reaction and fg of genomic DNA?

Response: We appreciate your interest on the analytical procedure for parasite load quantification. Parasite load was expressed as parasite genome equivalents per μg of tissue DNA. The equivalence between parasite genome equivalents and femtograms (fg) of DNA was calculated based on the estimated genome size of *Leishmania donovani*. The haploid genome of *L. donovani* is approximately 32 Mb, corresponding to about 3.5×10^{-14} g (≈ 35 fg) of DNA. Considering its diploid nature, the total genomic DNA content per parasite is approximately 70–100 fg; therefore, 1 parasite genome equivalent was considered equal to 100 fg of DNA.

Parasite DNA was quantified by qPCR using a standard curve generated from known concentrations of *L. donovani* genomic DNA (10^4 to 10^{-1} parasite equivalents per reaction). The parasite genome equivalents obtained from qPCR were then divided by the total DNA concentration of each sample (μg of tissue DNA measured by NanoDrop) to normalize for variation in DNA extraction yield. This provided the final result expressed as parasite genome equivalents per μg of tissue DNA. It has already been discussed in our previous article (reference: [doi: /10.1371/journal.pone.0185606](https://doi.org/10.1371/journal.pone.0185606)).

Comment: Could you specify which guide has been used to estimate intra-assay and inter-assay CVs?

Response: It has been discussed in our previous article.

(ref: [doi: /10.1371/journal.pone.0185606](https://doi.org/10.1371/journal.pone.0185606)).

Comment: Could you clarify the rationale for estimating parasitic loads based on the assumption of 10, 50, or 100 copies of the target sequence per *Leishmania* genome? Why not instead use the validated gene dosage information available from sequenced *Leishmania* genomes and apply the copy number already reported in GenBank?

Response: It has been discussed in the data analysis section in the line number 424-430.

It would have been more specific to determine the number of target genes from the complete and annotated genome sequences of Indian *Leishmania donovani* isolates. This has been mentioned as a limitation of the study in the discussion section.

Comment: Caption to Figure 2. Please clarify if the LOD exhibited in the standard curve is actually the limit of quantification... In general, LoD means LoD 95 and is calculated by performing a series of replicates along different working days, following the CLSI guides. The standard curve with six-dilution range (10,000 - 0.1 parasite/s per reaction) exhibiting the limit of detection (LOD) of ddPCR.

Response: You are absolutely correct. The elaboration of the LoD is "limit of detection". The standard curve with six-dilution range (10,000 - 0.1 parasite/s per reaction) exhibiting the limit of detection.

Comment: When you express sensitivity values lower than 1 parasite per reaction, it should be expressed as parasite genome equivalents/reaction.

Response: We really appreciate your critical observation. It has been revised accordingly in the manuscript.

Minor Revision

Comment: Line 165. The CV values demonstrate a high reproducibility in all the but the samples with very low DNA copies.

Response: It has been revised accordingly in the manuscript.

Comment: Table 1-Please replace (n%) by n (%)

Response: It has been revised accordingly in the manuscript.

Re: Spectrum01800-25R1 (**Application of ddPCR for diagnosis and treatment monitoring of visceral leishmaniasis patients: addition of an ultrasensitive tool to the diagnostic arsenal**)

Dear Dr. Prakash Ghosh:

Your manuscript has been accepted, and I am forwarding it to the ASM production staff for publication. Your paper will first be checked to make sure all elements meet the technical requirements. ASM staff will contact you if anything needs to be revised before copyediting and production can begin. Otherwise, you will be notified when your proofs are ready to be viewed.

Sincerely,
William Witola
Editor
Microbiology Spectrum

Reviewer #2 (Comments for the Author):

The authors have followed the comments of the reviewers, which has improved the clarity of the data obtained reported in the text and Tables, and in turn these changes improved the quality of the manuscript.